# The Use of ^18^F-Fluoride Positron Emission Tomography/Computed Tomography Scanning to Identify Sources of Pain after Posterior Lumbar Interbody Fusion—An Analysis in Patients with and without Symptoms

**DOI:** 10.3390/diagnostics14131327

**Published:** 2024-06-22

**Authors:** Marloes J. M. Peters, Boudewijn T. Brans, Wouter A. M. Broos, Elisabeth M. C. Jutten, Felix M. Mottaghy, Olaf Schijns, René E. Weijers, Paul C. Willems

**Affiliations:** 1Department of Orthopaedic Surgery, Maastricht University Medical Center, P.O. Box 5800, 6202 AZ Maastricht, The Netherlands; marloes.peters@maastrichtuniversity.nl (M.J.M.P.); l.jutten@mumc.nl (E.M.C.J.); 2Department of Nuclear Medicine, Albert Schweitzer Ziekenhuis, 3318 AT Dordrecht, The Netherlands; boudewijnbrans@gmail.com (B.T.B.); wouterbroos@gmail.com (W.A.M.B.); 3Department of Nuclear Medicine and Radiology, Maastricht University Medical Center, P.O. Box 5800, 6202 AZ Maastricht, The Netherlands; felix.mottaghy@mumc.nl (F.M.M.); r.weijers@mumc.nl (R.E.W.); 4Department of Nuclear Medicine, University Hospital RWTH Aachen, 52074 Aachen, Germany; 5Department of Neurosurgery, Maastricht University Medical Center, P.O. Box 5800, 6202 AZ Maastricht, The Netherlands; o.schijns@mumc.nl

**Keywords:** ^18^F-fluoride PET/CT, lumbar spinal fusion, pain

## Abstract

Background: Identifying the cause of recurrent or persisting pain after posterior lumbar interbody fusion (PLIF) is essential for establishing optimal treatment. In this study, we evaluate patients after PLIF surgery by ^18^F-fluoride PET/CT scans and patient-reported outcome measures (PROMs). Methods: A total of 36 PLIF patients were included. Sixty minutes after intravenous injection of ^18^F-fluoride, PET/CT scanning was performed. Bone graft ingrowth, subsidence, screw loosening and damage of facet joints were scored by quantifying the level of bone metabolism of the vertebral endplates in the disc spaces, around screws and around the facet joints on the PET scans. Results: In contrast to asymptomatic patients, symptomatic patients showed abnormal PET values around pedicle screws and/or facet joints and at the lower endplates of the disc spaces, identifying a possible source of pain. On CT, no significant differences between these two groups were found. Conclusion: The PET/CT findings appeared to correlate better with symptoms on PROMs compared to CT findings alone. When interpreting ^18^F-fluoride PET/CT findings after PLIF surgery, one should realize bone metabolism in the disc spaces of the operated segments and around pedicle screws or facet joint changes during follow-up, reflecting natural recovery.

## 1. Introduction

Lumbar interbody fusion aims to fix one or more vertebral segments using metal implants and bone grafts to treat patients suffering from low back pain irresponsive to conservative treatment. Pain is assumed to be relieved by stabilization of the segments, decompression of impinged nerve roots and creation of definite bony fusion of the intended segments. Unfortunately, 5–50% of fusion patients report recurrent or persisting back and/or leg pain after the operation [1]. Identifying the cause of persisting symptoms several months or even years after surgery is crucial for choosing the treatment strategy that will benefit the patient most. Persisting symptoms can have multiple causes [2] and pain that appears after surgery is not necessarily related to the operation itself [3], hampering straightforward diagnosis. Persisting or recurrent pain or disability can be assessed by physical examination or expressed by PROMs (patient-reported outcome measures). However, before (minimal) invasive treatment can be considered, imaging is necessary for exact identification of the presumed pain source [2] to increase the likelihood of successful treatment. 

At present, for imaging of patients with pain or disability after fusion surgery, plain radiographs can be used for the assessment of instrumentation and implants, magnetic resonance imaging (MRI) can be used to identify neurological compromise, or computed tomography (CT) can be used to rule out or confirm nonunion, subsidence or malposition of instrumentation. Unfortunately, there is seldom a strong relation between these findings and the symptoms of the patient [3,4,5]. Therefore, several nuclear-based modalities of imaging have been used to study bone metabolism in the operated segments of patients with persisting pain after spinal fusion.

In several reports, ^18^F-fluoride positron emission tomography (PET/CT) has been used to study patients with persisting pain or disability after spinal fusion [6,7,8,9,10,11,12]. With PET/CT scanning using bone-seeking tracer ^18^F-fluoride, local bone metabolism anywhere in the body can be localized and quantified [13]. The high diagnostic sensitivity of ^18^F-fluoride PET to identify the source of pain has been shown in back pain patients [14,15,16] or patients with otherwise unexplained bone pain [17,18,19,20]. However, for diagnostic specificity, hot spots in bone metabolism physiologically occurring after surgery should be distinguished from increased metabolism due to symptoms. For example, both continuing bone repair after successful fusion [21] and altered distribution of mechanical loads in adjacent segments [22] will show altered bone metabolism, although these are often not related to persisting symptoms. In order to better distinguish natural from pathologic recovery on PET imaging, we should analyze both symptomatic and asymptomatic patients after fusion surgery.

The goal of the present paper was to study the relation between findings on ^18^F-fluoride PET/CT and symptoms, as listed by PROMs (patient-reported outcome measures) in patients after PLIF surgery. The spinal structures of both the adjacent and operated segments were assessed by PET and CT. As such, the correlation of imaging findings and PROMs may identify possible pain indicators. The findings of patients without pain or disability were necessary to better interpret bone metabolism changes inherent to the fusion surgery versus changes indicative of a possible pain source.

## 2. Materials and Methods

### 2.1. Patients

Between June 2008 and April 2015, 40 patients consented to undergo a postoperative PET/CT scan after PLIF surgery. In total, 36 of these patients had filled out questionnaires and could subsequently be included in the current retrospective study. Of these patients, 16 complained of recurrent or persistent back pain for which no evident cause had been found with conventional imaging. Therefore, these patients had undergone a PET/CT scan for more diagnostic information at variable lengths of follow-up and retrospectively were included in this study. The other 20 of the 36 patients included could be recruited from a prospective cohort who underwent standard control by means of clinical examination and imaging (standard radiographs, PET/CT) at 1-year follow-up after PLIF to evaluate the first postoperative year. The operative technique of PLIF has been described earlier [9,23]. Three patients had been fixed at two levels. The levels operated on were L5-S1 (n = 21), L4-L5 (n = 15) and L3-L4 (n = 3). For the whole group, PET/CT scans had been executed at a mean follow-up of 18.8 months (median 12.5, range 9–76 months). For the remainder of the study, the 36 included patients were analyzed as a whole, irrespective of whether inclusion was retrospective or prospective. The current study was performed according to the Helsinki Declaration of 1975, as revised in 2013, and its protocol was approved by the medical ethical committee of Maastricht UMC (NL.32881.068.11). From all patients, written informed consent was obtained.

### 2.2. Procedure of ^18^F-Fluoride PET/CT

A median of 196.5 MBq (100–248 MBq, mean 194.4) of ^18^F-fluoride was injected intravenously 60 min before images were obtained with an integrated PET/CT scanner (Gemini TF 64 PET-CT, Philips, Eindhoven, The Netherlands). For localization, a CT scan from the lumbosacral spine was acquired (120 kV, 30 mAs, slice thickness 4 mm), after which a 5 min static PET scan of two bed positions each was performed. Directly after, a diagnostic CT scan (64-slice helical, 120 kV, 250 mAs, slice thickness 1 mm, increment 0.8 mm) was obtained and the scans could then be viewed on clinical software (EBW, Philips, The Netherlands) and analyzed further on the PMOD 3.0 research tool (PMOD Technologies Ltd., Zürich, Switzerland).

### 2.3. Analysis of Diagnostic CT Scans

The diagnostic CT scans were evaluated by three blinded observers independently: a musculoskeletal radiologist with >20 years of experience (RW), a nuclear medicine physician with >15 years of experience (BB) and a junior nuclear medicine physician (WB). Discrepancies were resolved between the observers by consensus. Bony fusion within the disc space (CT_FUSION) was scored at all operated segments as 0, 1 or 2, depending on the amount of osseous bridging between the involved vertebrae (Figure 1A) [9]. When bone mass progressed from one endplate to the other side of the disk space without interruption, it was defined as bony or osseous bridging. In patients who had been operated on at two segments, the segment with the lowest fusion score was taken (worst-case scenario). Next, the severity of subsidence of the intervertebral cages (CT_SUBS) into the endplates was scored as 0, 1 or 2 (Figure 1B) [9]. The vertebral endplate was visualized and outlined, and as such the degree of subsidence into the vertebral body could be determined. Thus, subsidence scores for both lower and upper endplates could be scored for all operated segments. The presence or absence of radiolucent lines around the pedicle screws was used to score screw loosening (CT_SCREW). All screws were defined as 0 (no signs of loosening) or 1 (radiolucency) (Figure 1C). In case of 1 loose screw, this was categorized as 1 (signs of loosening). For surgical access to the disc space, facet joints of the operated levels had been removed. Of the two adjacent levels above the fusion, the facet joints (CT_FACET) were scored as normal (1) or degenerated (2) (Figure 1D). If only one facet joint was degenerated, the patient’s facet joints were categorized as degenerated.

### 2.4. Analysis of ^18^F-Fluoride PET Scans

Three blinded observers (MP, BB, WB) evaluated the PET scans independently. If there were any discrepancies between the observers, these were resolved by consensus. After drawing volumes of interest (VOIs) in the low-dose CT images, these were transferred to the co-registered ^18^F-fluoride PET images and thus the maximum standardized uptake values of ^18^F-fluoride (SUVmax) could be calculated as a quantification of bone metabolism. Bone metabolism was calculated in the disc spaces and at the endplates for all operated segments. In every operated segment, 3 ellipsoid-shaped VOIs were drawn manually on the low-dose CT images by which the vertebral contours were followed (slice thickness 4 mm, short axis range: 40–50 mm, long axis range: 55–65 mm) in order to measure SUVmax at the upper endplate (SUVmax_endUP), the lower endplate (SUVmax_endLOW) and in the disc space (SUVmax_inter) (Figure 2A), as outlined previously [9]. By drawing a rectangular beam (dimensions 50 × 20 × 20 mm) on low-dose CT around each screw, the bone metabolism around the pedicle screws could be calculated (Figure 2B). For further analysis, the highest screw activity within a patient (SUVmax_screw) was used. For measuring facet joint metabolic activity, 10 mm diameter spheres were drawn around the joints on CT, after which SUVmax could be determined for every individual facet joint (Figure 2C). For analysis in every patient, the facet joint with the highest SUVmax was used (SUVmax_facet).

### 2.5. Patient-Reported Outcome Measures (PROMs)

The amount of pain and disability per patient was quantified by means of three validated questionnaires commonly used in patients with back pain:The Oswestry Disability Index (ODI) evaluates ten general daily tasks based on back-related function [24] and can be used to assess functional outcome in patients treated for lower back pain [25].Back pain and/or leg pain was quantified by Visual Analogue Scale (VAS) [26]. Patients were asked to express the amount of pain in their back, right and left leg ranging from 0 (no pain) to 100 (worst pain). The score for back pain was used as a measure in this study.The EuroQol (EQ)-5D measures health-related quality of life in five domains (i.e., mobility, self-care, usual activities, pain/discomfort and anxiety/depression) [27]. The EQ-5D index score was calculated based on a Dutch value set, representative of the Dutch population with regard to age and gender [28].

All questionnaire scores were linearly re-scaled from 0 (worst possible score) to 100 (best possible score) to facilitate further analysis. The 36 patients included were divided into categories based on these re-scaled scores; patients with scores 0–40 were classified as category 1 (“worst score”), scores 40–60 as category 2 (“intermediate”) and scores 60–100 category 3 (“best score”), respectively (regardless of whether patients were included retrospectively or prospectively). Patients were defined as ‘asymptomatic’ if all 3 questionnaire scores were >60 and at least 2 scores >80. The bone metabolism values of those patients considered ‘asymptomatic’ were averaged for all spinal structures involved in order to obtain a map with ‘successful recovery’ bone metabolism values of spinal structures after PLIF surgery.

### 2.6. Statistical Evaluation

For statistical analysis, IBM SPSS Statistics 23.0 (Armonk, NY, USA: IBM Corporation) was used. All 36 patients were divided into categories based on PROM scores as well as on CT scores. Differences between these categories were tested by an independent t-test in case of normality and by a Mann–Whitney U-test (2 categories) or a Kruskal-Wallis test (>2 categories) otherwise. If the Kruskal-Wallis test showed significant differences between categories, the Mann–Whitney U-test was used for specification of those categories between which these differences were present. *p*-values ≤ 0.05 were defined as a statistically significant difference.

## 3. Results

### 3.1. Relation between PROMs and CT Findings

According to the four CT parameters (facet joints, screw loosening, subsidence and fusion), all patients were categorized. In Table 1, the mean values of PROM scores are listed per CT category. There were no significant differences in PROM scores between different CT-based categories.

### 3.2. PET Findings Versus PROM Scores

All patients were categorized as based on their scores on the ODI, EQ-5D and VAS for pain. In Table 2, the SUV values for bone metabolism are listed per PROM category for the upper and lower endplates and the disc spaces of operated segments, as well as for facet joints and pedicle screws. Between these categories, significant differences in bone metabolism were found for the lower endplates, facet joints, and around the pedicle screws, as illustrated in Figure 3. In patients with the worst ODI scores (category 1), higher values of bone metabolism were found at the lower endplates, around the pedicle screws and at the facet joints as compared to patients with better ODI scores. This was similar for the EQ-5D questionnaire. Pain on VAS correlated with bone metabolic activity surrounding the screws.

### 3.3. PET Findings Versus CT Findings

Correlations between bone metabolic activity on PET and anatomical findings on CT were determined. For this, all patients were categorized as based on the results of the four CT parameters (i.e., subsidence, fusion, facet joint status and screw loosening). In Table 3, the distribution of bone metabolism (mean ± sd) of both upper and lower endplates, disc spaces, facet joints and screw surroundings per CT category is shown, including *p*-values for differences. In Figure A1 in Appendix A, a grid of figures that correlate each PET score to each CT category is provided. For both endplates, the amount of bone bridging on CT was inversely proportional to bone metabolism on PET. Additionally, subsidence as measured on CT was related to the PET metabolic activity of both the adjacent facet joints and the lower endplates. Higher bone metabolism values were found in degenerated facets as compared to normal adjacent facet joints.

### 3.4. Bone Metabolism Profile on PET after Successful Spinal Fusion

In total, 14 patients were considered ‘asymptomatic’ based on their PROM scores (see criteria specified in Methods). In these patients, bone metabolic activity in each spinal structure was averaged in order to obtain a “normal” bone metabolism profile after PLIF surgery (Figure 4). Typically, ‘asymptomatic’ patients after PLIF showed the following: 1. much higher activity at both endplates of the operated segment than adjacent non-operated levels; 2. comparable PET activity (SUVmax) in the middle of the disc space versus at the endplates of operated segments, whereas in non-operated levels, intervertebral PET activity was much less than activity at the endplates; 3. only moderately elevated PET activity of the facet joints one level above the fusion; and 4. minimal bone metabolic activity around the pedicle screws.

## 4. Discussion

The main finding of the current study on the use of ^18^F-fluoride PET/CT scanning in patients after PLIF surgery is that local ^18^F-fluoride PET activity appeared to correlate better to clinical outcome (as assessed by PROMs) than CT parameters did. Patients with persisting symptoms displayed higher bone metabolic activity at the lower endplates, the adjacent facet joints and around the pedicle screws at the minimum follow-up of 1 year. However, not all structures with increased PET values should be considered pain generators; regardless, ‘asymptomatic’ patients after PLIF were characterized by the following: 1. much higher activity at both endplates of the operated segment than adjacent non-operated levels; 2. comparable PET activity in the middle of the disc space versus at the endplates of operated segments, whereas in non-operated levels intervertebral PET activity was considerably less than at the endplates; 3. moderately elevated PET activity of the facet joints one level above the fusion; and 4. minimal bone metabolic activity around the pedicle screws. So, these bone metabolism patterns are inherent to the operative procedure and should be regarded as normal course after PLIF.

Another interesting finding of this study is the functional interpretation of elevated PET findings in spinal structures as possible sources of persisting pain and disability after PLIF surgery. Patients with persisting symptoms showed abnormal PET activity at the lower endplates, around the pedicle screws, and at adjacent facet joints, which may resemble an aberrant redistribution of mechanical stresses to these structures. This transfer of stresses was also shown by subsidence on CT, which appeared to correlate well with higher PET metabolic activity at the lower endplates. This would imply that subsidence may cause aberrant high bone stresses at the vertebral endplates. Moreover, subsidence appeared to be correlated with higher PET activity of the facet joints. As the operated segment decreases in height, subsidence may increase tension on the facet joints above, which was illustrated by higher PET activity of adjacent facet joints. 

The use of ^18^F-fluoride PET/CT scanning has been explored earlier in patients with persisting pain and disability following spinal fusion surgery when no obvious clinical or conventional imaging findings could explain these symptoms. In a study of eight patients with postoperative back pain after lumbar fusion and with negative imaging findings, Gamie et al. [8] analyzed the uptake of ^18^F-fluoride in adjacent facet joints and operated intervertebral disc spaces and identified positive areas of uptake in all eight patients. Fischer et al. [7] reported on 15 patients with persisting pain and disability at 1–10 years follow-up after spinal fusion. In this study, PET values higher than those of normal surrounding bone were interpreted as pathologic and as such aberrant increased uptake of tracer was observed in 8 of the 17 intervertebral cages. It was concluded that the uptake of ^18^F-fluoride tracer was increased in half of the cages, which may indicate aberrant stress transfer of micro instability leading to unsuccessful fusion. In a study by Quon et al. [11], PET/CT findings were evaluated at 4–96 months follow-up in 22 patients suffering from recurrent or persistent symptoms after fusion, which could not be explained by physical examination or CT scan. Based on abnormal uptake of ^18^F-fluoride tracer at the level of grafts, cages, rods or screws, revision surgery was performed in 15/22 patients. It appeared that the presence of aberrant findings at revision surgery such as graft fracture, cage failure, or screw loosening had been predicted correctly by PET/CT scanning in 14/15 patients. In a study of Seifen et al. [12], a total of 334 pedicle screws were evaluated in 59 patients by PET/CT scan. In this study, the SUVmax surrounding each pedicle screw was calculated in order to assess screw loosening, after which the final diagnosis was made by either surgical exploration (27 patients) or clinical follow-up (32 patients). It was found that in those 58/334 screws that appeared to be loose, the average SUVmax was 17.10 ± 6.6, whereas in those 276/334 screws that were not loose, the average SUVmax was significantly lower (14.10 ± 5.9). Thus, the calculated sensitivity and specificity of the ^18^F-fluoride PET/CT scan to predict screw loosening were 75% and 97.4% in the patient-based analysis and 45.6% and 100% in the screw-based analysis, respectively. It was concluded that ^18^F-fluoride PET/CT scanning can identify screw loosening and resulting implant instability with high accuracy [12]. In a study of Pouldar et al. [10], it was shown that the use of PET/CT scanning helped to identify sources of pain in 17 of the 25 patients with persisting symptoms at 15 months of follow-up after spinal fusion, in which pain was verified by findings at surgical revision and/or facet joint blocks or the results of non-invasive treatment. The authors stated that PET/CT scanning was better in identifying underlying pathology in postop patients than CT scans alone, specifically when detecting screw loosening or nonunion.

One of the strengths of the current study is that, unlike in the abovementioned studies, both patients with and without persisting symptoms were included for ^18^F-fluoride PET/CT analysis, which enabled us to make a better distinction between PET findings in patients with a natural ‘asymptomatic’ course and PET findings in those with an aberrant ‘pathological’ course after PLIF. PROMs were used as an acknowledged and validated means to quantify the severity of pain and disability. Furthermore, reproducible and objectively quantifiable methods were used for analysis of PET/CT and CT findings in order to minimize bias of subjective individual interpretation by a single radiologist or physician in nuclear medicine only. A limitation, however, of the current explorative study is that the ^18^F-fluoride PET/CT findings were not used for treatment strategy, meaning that we gathered data on the diagnostic accuracy of the PET/CT information but not on the prognostic accuracy of PET/CT, which could be used for surgical decision making. Therefore, we plan a follow-up study in which treatment is based on PET/CT findings in patients with persisting symptoms after PLIF. E.g., high PET activity at adjacent facet joints would lead to facet joint blocks and/or adjacent fusion, and high activity at the vertebral endplates would indicate revision surgery of the index segment. Subsequently, observations at revision surgery could confirm the PET/CT findings. Next, although the total number of patients in this study was acceptable, the division into subgroups may have resulted in small categories of patients, thus limiting the findings of this study. As such, only two patients with signs of screw loosening were observed, making it hazardous to draw conclusions in contrast to the study of Seifen et al., in which 58 of 334 screws showed signs of loosening [12]. Additionally, there was variety in the length of follow-up (time period between PLIF and evaluation by PET/CT) between patients. Variability in bone metabolic activity may occur in different stages of bone repair, which may have biased the results of our study.

Currently, few hospitals have access to ^18^F-fluoride PET/CT scans. However, more widely available nuclear imaging techniques, e.g., technetium bone scanning (SPECT/CT), may serve well as alternative tools [29]. Recent developments in SPECT/CT allow for quantitative whole-body imaging of radiopharmaceutical distribution, similar to PET/CT, albeit with longer acquisition times [30,31]. Because of the close relation with symptomatology, a PET/CT or SPECT/CT scan may be considered when routine diagnostic procedures are inconclusive, which may be deemed proportional in relation to the individual burden and high overall societal costs of patients with persisting pain and disability after spinal fusion. For patients suffering from spinal complaints, it may be painful and difficult to lay on their back in the scanner during acquisition. New developments in long axial field-of-view PET/CT scans might be beneficial for this patient group as this allows for faster scan acquisition [32].

In conclusion, the PET/CT findings appeared to correlate better with the clinical status as assessed by PROMs than findings on CT alone did. Further research is needed to verify if a treatment strategy based on PET/CT findings will provide a better outcome for patients with persisting symptoms after spinal fusion surgery. For the interpretation of ^18^F-fluoride PET/CT scans, it is crucial to realize that an uneventful surgical procedure itself or length of follow-up will change the pattern of bone metabolism in and around operated segments. Therefore, changes in uptake of ^18^F-fluoride on PET/CT may be classified as ‘pathological’ only when clearly aberrant from the uptake patterns of patients with a natural ‘asymptomatic’ course after spinal fusion.

## Figures and Tables

**Figure 1 diagnostics-14-01327-f001:**
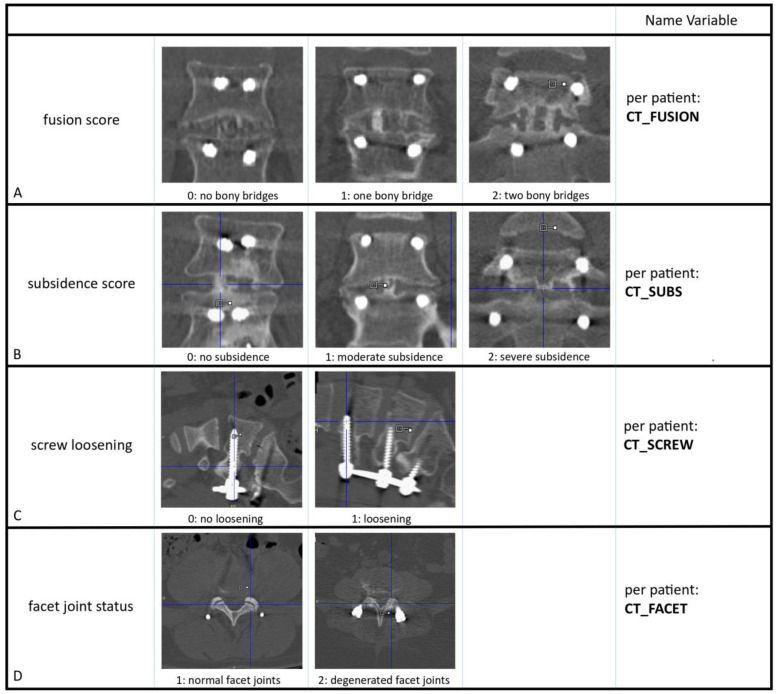
Overview of the variables as scored on CT for every patient. (**A**): CT_FUSION: Fusion was categorized as 0 (absence of bone bridges), 1 (one bone bridge) or 2 (at least 2 bone bridges). (**B**): CT_SUBS: Subsidence was categorized as no subsidence (0), moderate (1) or severe subsidence (2). (**C**): CT_SCREW: Screw loosening was judged by the experienced observers as absent (0) or present (1). (**D**): CT_FACET: Facet joint status was categorized as normal (1) and degenerated (2).

**Figure 2 diagnostics-14-01327-f002:**
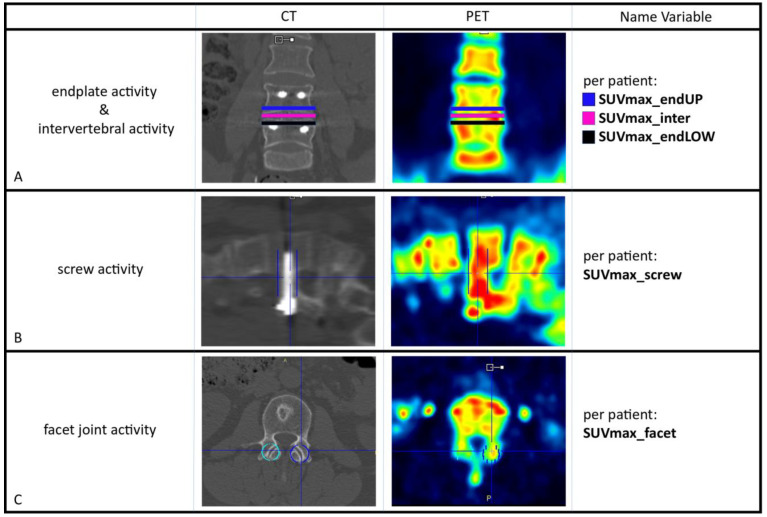
The volume of interest (VOI)-based method was used to calculate SUVmax for the spinal structures involved with the corresponding names of the variables for every patient. The VOIs obtained from the low-dose CT image were transferred to the co-registered PET images in order to calculate SUVmax. (**A**): For every operated segment, 3 VOIs were drawn (at both endplates and in the intervertebral disc space) for which SUVmax_endUP, SUVmax_endLOW and SUVmax_inter could be calculated, respectively. (**B**): Around the pedicle screws, a rectangular beam was drawn. For the individual patient, the screw showing the highest SUVmax was used for analysis (SUVmax_screw). (**C**): Around the facet joints, spheres were drawn (depicted in light blue and dark blue in the figure), and for every patient, the joint revealing the highest SUVmax was used for analysis (SUVmax_facet).

**Figure 3 diagnostics-14-01327-f003:**
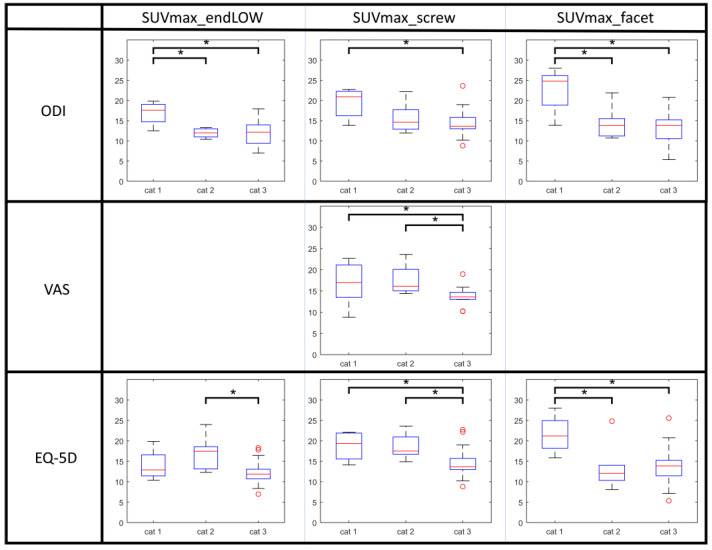
Display of all correlations between PROM categories and PET values that were significant. Patients have been categorized based on questionnaire scores: those with scores 0–40 were categorized as cat. 1 (“worst score”), those scoring 40–60 as cat. 2 (“intermediate”), and those scoring 60–100 as cat. 3 (“best score”), respectively. Significant differences in PET scores between patient categories are marked by asterisks (*p* < 0.05).

**Figure 4 diagnostics-14-01327-f004:**
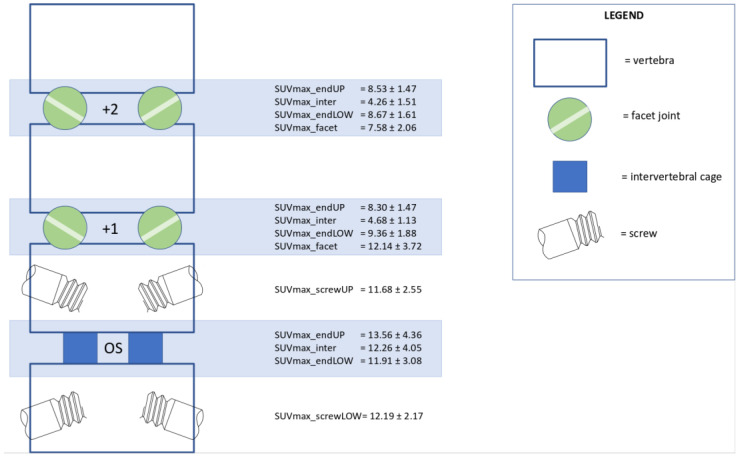
In this schematic drawing, the bottom segment at the left resembles the operated segment (OS) of the lower spine with the intervertebral cage and pedicle screws. Above that, adjacent vertebrae and facet joints are drawn. The mean PET values of bone activity ± standard deviation of the 14 ‘asymptomatic’ patients are listed per structure in the middle column. The legend at the right illustrates the drawings that explain each spinal structure.

**Table 1 diagnostics-14-01327-t001:** Mean values of the PROM questionnaires, ODI, VAS and EQ-5D, for the CT-based categories of facet joint status, screw loosening, subsidence, and fusion. For each CT category, the number of patients is reported. The *p*-values of differences in PROM scores for the different CT-based categories were assessed by a Kruskal–Wallis test if there were 3 CT categories and by a Mann–Whitney U-test if there were only 2 CT categories. The subsidence of four patients and the screw loosening of two patients could not be scored due to artefacts in the specific area on CT caused by the inserted hardware during surgery (i.e., screws and rods to provide primary stabilization).

			PROMs		
CT Parameter	Score	Number of Patients	ODI [Mean ± Stdev]	VAS [Mean ± Stdev]	EQ-5D [Mean ± Stdev]
fusion	0	14	61.57 ± 25.39	47.86 ± 25.17	66.57 ± 26.21
	1	9	85.71 ± 14.40	71.43 ± 24.28	86.13 ± 7.97
	2	13	61.79 ± 24.48	49.23 ± 32.59	74.00 ± 21.46
	*p*-value		0.052	0.164	0.179
subsidence	0	6	59.45 ± 18.06	40.83 ± 28.00	72.85 ± 14.29
	1	20	62.83 ± 27.31	52.50 ± 32.79	68.96 ± 28.19
	2	6	73.00 ± 21.19	52.50 ± 15.41	72.73 ± 23.52
	*p*-value		0.597	0.624	0.829
screw loosening	0	32	62.92 ± 24.02	48.75 ± 29.65	70.70 ± 23.71
	1	2	82.00 ± 16.97	55.00 ± 21.21	70.51 ± 41.71
	*p*-value		0.321	0.699	0.856
facet joint status	1	21	69.97 ± 22.58	54.52 ± 28.54	73.70 ± 21.75
	0	15	59.73 ± 26.01	46.33 ± 30.91	68.88 ± 26.89
	*p*-value		0.309	0.465	0.800

**Table 2 diagnostics-14-01327-t002:** Mean ± standard deviation of the five PET parameters (SUVmax of the upper endplate, intervertebral disc space, lower endplate, screws and facet joints) for the categories of the three PROMs (ODI, VAS and EQ-5D). Patients with scores 0–40 were classified into category 1 (“worst score”), patients with scores 40–60 into category 2 (“intermediate score”) and patients with scores 60–100 into category 3 (“best score”), respectively. For the PROM categories, differences in PET scores were analyzed by the Kruskal–Wallis test, for which *p*-values < 0.05 are indicated by a red font.

			PET Parameter				
PROMs	Category	Number of Patients	EndUP [Mean ± Stdev]	Inter [Mean ± Stdev]	EndLOW [Mean ± Stdev]	Screw [mean ± Stdev]	Facet Joint [Mean ± Stdev]
ODI	1	6	13.50 ± 4.83	12.51 ± 4.14	16.89 ± 3.15	19.32 ± 3.79	22.54 ± 5.56
	2	10	14.01 ± 1.70	15.27 ± 3.83	11.93 ± 1.09	15.96 ± 3.56	13.96 ± 3.44
	3	20	13.98 ± 4.55	12.61 ± 4.56	12.23 ± 3.04	14.43 ± 3.24	13.27 ± 4.11
	*p*-value		0.806	0.189	0.046	0.049	0.024
VAS	1	17	13.54 ± 3.14	13.93 ± 4.32	13.44 ± 3.52	16.69 ± 4.19	15.65 ± 6.48
	2	4	16.37 ± 5.20	14.81 ± 6.32	12.99 ± 0.61	17.56 ± 4.13	16.18 ± 3.51
	3	15	13.69 ± 4.23	12.44 ± 3.96	11.94 ± 2.97	13.78 ± 2.13	13.41 ± 3.38
	*p*-value		0.572	0.740	0.430	0.038	0.594
EQ-5D	1	4	13.74 ± 3.06	12.37 ± 2.36	14.00 ± 4.08	18.72 ± 3.85	21.56 ± 5.01
	2	6	15.63 ± 4.84	17.56 ± 4.90	15.30 ± 2.81	18.52 ± 3.18	13.55 ± 5.88
	3	26	13.55 ± 3.74	12.63 ± 4.05	11.99 ± 2.76	14.41 ± 3.22	14.00 ± 4.29
	*p*-value		0.568	0.115	0.017	0.006	0.028

**Table 3 diagnostics-14-01327-t003:** Mean values of the PET activity of the upper and lower endplates, intervertebral disc space, pedicle screws and adjacent facet joints for each CT category. Per CT category, the number of patients is reported, and the *p*-values for statistical differences in PET activity between these categories are listed. *p*-values ≤ 0.05 are displayed in red. The subsidence of four patients and the screw loosening of two patients could not be scored due to artefacts in the specific area on CT caused by the inserted hardware during surgery (i.e., screws and rods to provide primary stabilization).

			PET Parameter				
CT Parameter	Category	Number of Patients	EndUP [Mean ± Stdev]	Inter [Mean ± Stdev]	EndLOW [Mean ± Stdev]	Screw [mean ± Stdev]	Facet Joint [Mean ± Stdev]
fusion	0	14	16.27 ± 4.21	14.34 ± 5.72	15.86 ± 3.68	16.94 ± 3.78	16.17 ± 5.93
	1	9	13.36 ± 2.65	13.61 ± 1.99	10.67 ± 1.59	14.00 ± 0.92	14.10 ± 2.17
	2	13	11.70 ± 3.07	12.64 ± 5.50	11.50 ± 2.68	14.91 ± 4.43	13.33 ± 5.52
	*p*-value		0.015	0.513	0.001	0.170	0.451
subsidence	0	6	12.32 ± 4.02	12.05 ± 6.36	10.56 ± 2.38	13.39 ± 4.77	9.96 ± 3.27
	1	20	13.73 ± 3.54	13.31 ± 3.71	13.27 ± 3.20	15.67 ± 3.22	15.64 ± 5.10
	2	6	15.77 ± 5.55	15.18 ± 8.07	16.34 ± 4.67	17.75 ± 4.51	15.85 ± 6.21
	*p*-value		0.551	0.920	0.039	0.129	0.038
screw loosening	0	32	13.84 ± 4.02	13.57 ± 5.12	13.13 ± 3.66	13.41 ± 3.46	15.02 ± 5.29
	1	2	13.88 ± 2.88	11.88 ± 1.73	15.05 ± 4.09	11.24 ± 3.15	10.89 ± 3.99
	*p*-value		0.856	0.471	0.556	0.077	0.257
facet joint status	1	21	13.39 ± 3.69	13.71 ± 5.71	12.07 ± 3.60	15.05 ± 3.70	8.92 ± 3.42
	2	15	14.71 ± 4.16	13.40 ± 4.61	14.47 ± 3.32	16.30 ± 3.72	12.17 ± 6.22
	*p*-value		0.309	0.590	0.007	0.545	0.001

## Data Availability

The raw data supporting the conclusions of this article will be made available by the authors on request.

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
