# Peer review of "The Use of 18F-Fluoride Positron Emission Tomography/Computed Tomography Scanning to Identify Sources of Pain after Posterior Lumbar Interbody Fusion—An Analysis in Patients with and without Symptoms"

_diagnostics, 2024, doi:10.3390/diagnostics14131327_

Round 1

Reviewer 1 Report

Comments and Suggestions for Authors

The work under evaluation is research conducted on the use of PET/CT with 18F-fluoride in patients undergoing surgical treatment (PLIF) and with persistent pain symptoms. The authors found a correlation, in patients with persistent pain, between the degree and pattern of uptake on PET/CT and the symptoms assessed through self-assessment questionnaires.

Introduction: well done and easy to read. Methods: well described. I would avoid detailing the definition of SUV, considering the readership. Results: clear and well described. Discussion: I would enrich the discussion with two short paragraphs. When the authors talk about the use of SPECT/CT, highlighting the limited use of PET with 18F-fluoride, I would add some considerations on the potential of quantitative whole-body SPECT/CT in this context, citing some works: doi: 10.3390/diagnostics13182971, doi: 10.1053/j.semnuclmed.2022.01.004. Additionally, I would add that in patients with lower back pain and difficulty maintaining a supine position, the new technology with the so-called long axial-field-of-view PET scanners can offer some advantages, such as the implementation of 'fast' or 'ultra-fast' protocols, citing the following works: doi: 10.3390/diagnostics12020426, doi: 10.1080/17434440.2022.2141111

Author Response

please find reply in the attachment 

Reviewer 2 Report

Comments and Suggestions for Authors

This study aimed to identify sources of pain following posterior lumbar interbody fusion using F-fluoride PET/CT scanning. The paper is well-written, highly informative, and addresses an interesting topic. The methodology is thorough, and the discussion section is well-documented. I understand that you are planning a follow-up study in which PET/CT findings will be utilized to inform treatment strategies. Best of luck with that endeavor. Regarding the current paper, please address the following issues I have identified:

In this study, 36 patients were included, with 39 levels operated on, as 3 of them underwent surgery at 2 levels. However, I have noticed a discrepancy between the number of patients analyzed in each section. I understand that for patients who underwent surgery at two segments, the segment with the lowest fusion score was considered. However, the numbers still do not align. Is there an explanation for this discrepancy? Initially, in the Materials and Methods section, it is stated that PET/CT scans were performed on the entire group.

Another issue I found is the discrepancy between the Methods and Materials section and the Results section for the successful spinal fusion group. In the Methods and Materials section, it is stated that 20 of the 36 included patients underwent a standard control at a 1-year follow-up after PLIF to evaluate normal recovery. However, the Results section indicates that 14 patients had a successful outcome. Could you provide more specificity in the methodology regarding the criteria and number of patients for the "successful" group?

Quality of images in Figure 3 is not adequate. Please provide superior resolution images in order to assess the content of the Figure.

Please rework the tables to comply with the instructions for authors and template of the journal.

Respectfully,

Comments on the Quality of English Language

English is fine, only requires several minor edits.

Author Response

please find reply in the attachment

Round 2

Reviewer 2 Report

Comments and Suggestions for Authors

The revisions brought forth by the authors have improved the quality of the manuscript.

Comments on the Quality of English Language

Only few minor grammar and phrasing issues.